# Modified Modulation Strategy of 1: 1: 2 Asymmetric Nine-Level Inverter and Its Power Balance Method

**Manyuan Ye \*, Qiwen Wei, Wei Ren and Guizhi Song**

College of Electrical and Automation Engineering, East China Jiaotong University, Nanchang 330013, China; qwwei@ecjtu.edu.cn (Q.W.); rwei@ecjtu.edu.cn (W.R.); gzsong@ecjtu.edu.cn (G.S.)

**\*** Correspondence: myye@ecjtu.edu.cn

**Abstract:** The three unit nine-level inverter can output more voltage levels with fewer h-bridge units, while having better output waveform quality. However, in the conventional hybrid frequency modulation strategy, only one low-voltage unit adopts pulse width modulation (PWM), which causes the problem of switching loss and uneven heat distribution between the two low-voltage units. At the same time, the output power of the conventional modulation strategy is unbalanced. Aiming to resolve the above problems, a modified hybrid modulation strategy and a power balance control method under the strategy is proposed in this paper. The modulation strategy achieves output power balance between the three units and an even distribution of switching losses between the two low voltage units while maintaining the same output power quality. Simulation and experimental results verify the feasibility of the modulation strategy.

**Keywords:** multilevel inverter; modulation strategy; asymmetric cascade; linear balance

## 1. Introduction

The cascaded multi-level inverter becomes one of the most widely used inverter topologies in medium voltage high power transmission system for it can output a higher level of voltage and a higher number of levels without increasing the withstand voltage of the device. [1,2]. Cascaded H-bridge (CHB) multi-level inverters have been widely used in photovoltaic grid-connected, electric vehicles, and high voltage direct current transmission due to their small d$v$/d$t$, phase voltage redundancy, low harmonic component and easy modularization [3–5]. The CHB type multilevel inverter topology can be divided into type I, type II, and type III according to the DC side voltage level of each unit. Compared with the type I topology, type II and type III can output more voltage levels with fewer units [6]. For a CHB inverter, the DC-side voltage and switching frequency of each H-bridge unit can be different, and each unit can adopt a different modulation strategy, so it can achieve better output characteristics [7–11].

For asymmetric cascaded multilevel inverter topologies with different DC power supply voltage levels, it is difficult to output an ideal multilevel voltage waveform only by the modulation strategy of phase shift PWM (PS-PWM) [12] or phase disposition PWM (PD-PWM) [13]. Hybrid frequency PWM (HF-PWM) technology has emerged at the historic moment, which can enable switching devices of different voltage levels to work together in the same topology, so the characteristics of each switching device can be fully utilized.

However, for a cascaded multilevel inverter, the output power of each unit is usually unbalanced. When using a battery to supply power, this problem will cause the battery to discharge unevenly and affect the output power quality [14,15]. Therefore, it is important to ensure that the output power of each unit of the cascaded multilevel inverter is balanced. A modified carrier disposition modulation strategy (MCD-PWM) is provided for the power balance problem of the three-unit cascaded seven-level

inverter with the DC voltage ratio 1:1:1 in the [16], to enable the output power of each cascaded unit to be equal with each other. Compared with the traditional PS-PWM, the harmonic component of the line voltage and line current for the inverter output under the modified CD-PWM strategy is less, but the operating frequency of the switching devices is higher than that under the CPS-SPWM strategy, and the switching loss increases. The level comparison method used in [17] has a simple modulation strategy and a very low loss. However, when the modulation depth is lower than 0.25, the three H-bridge units have no voltage output, the fundamental of output voltage and modulation depth are nonlinear. It is not easy to accurately control the fundamental amplitude of the output voltage. The hybrid modulation strategy of [18] can achieve continuous PWM modulation of the output voltage, but the modulation strategy using only PWM modulates one low-voltage unit, while the other two units still use the level comparison method to output low-frequency square waves. The distribution of switching times is uneven in one cycle between two low-voltage units, which causes a difference in the heating conditions of the two units so that the low-voltage unit that outputs the high-frequency PWM is more likely to be damaged by heat.

Aiming to address the problem of output power unbalance in asymmetric nine-level inverters, a modified hybrid frequency pulse width modulation (MHF-PWM) strategy is proposed in this paper based on the traditional HF-PWM modulation strategy. The output power is balanced between the two low voltage units. Then the relationship between the output power and the modulation depth of each unit under the control of MHF-PWM modulation strategy is analyzed. Based on this, the power balance control method of MHF-PWM modulation strategy is proposed. The output power balance control of the inverter unit at full modulation depth is realized.

## 2. Topology of Asymmetric Nine-Level Inverter

Asymmetric nine-level inverter topology is shown in Figure 1 which is cascaded from three H-bridge units. The DC side voltages of the three H-bridge units H1, H2, and H3 are $2E$, $E$, and $E$, respectively. The output voltages of the three units are defined as $u_{H1}$, $u_{H2}$, and $u_{H3}$, the output phase voltage is $u_{AN}$, the fundamental amplitude of the output phase voltage is $u_{AN(1)}$, and the output phase current is $i_o$.

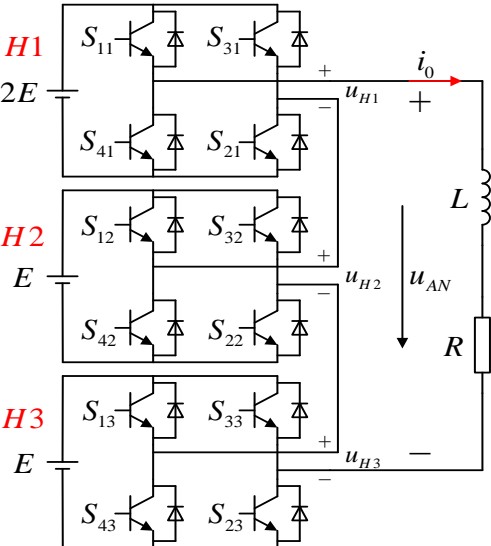

**Figure 1.** Topology of asymmetric nine-level inverter.

Define the switching function $S_i$, $i$ is the H-bridge unit number, and the value of the switching function $S_i$ is as follows

$$S_i = \begin{cases} 1 & S_{1i}, S_{2i} \text{ on} \\ 0 & S_{1i}, S_{3i}/S_{4i}, S_{2i} \text{ on} \\ -1 & S_{3i}, S_{4i} \text{ on} \end{cases} \tag{1}$$

The output phase voltage $u_{AN}$ can be expressed as Equation (2)

$$u_{AN} = S_1 \times 2E + (S_2 + S_3) \times E \tag{2}$$

According to formulas (1) and (2), there are three kinds of switching states in one H-bridge unit, and there are a total of 27 switching states in the three-unit combination. In order to prevent current inrush between H-bridge units, the following provisions are made: When the output voltage of the inverter is positive, all cascading units can only output positive level or 0; when the output voltage of the inverter is negative, all cascading units can only output a negative level or 0. The switching function that satisfies the above conditions is shown in Table 1. It can be seen that the inverter has redundant switching states at the outputs $\pm E$, $\pm 2E$, $\pm 3E$.

**Table 1.** Inverter output voltage and switching function.

| $u_{AN}$ | $S_1\ S_2\ S_3$ | $u_{AN}$ | $S_1\ S_2\ S_3$ |
|---|---|---|---|
| 4E | 1 1 1 | −4E | −1 −1 −1 |
| 3E | 1 1 0, 1 0 1 | −3E | −1 −1 0, −1 0 −1 |
| 2E | 1 0 0, 0 1 1 | −2E | −1 0 0, 0 −1 −1 |
| E | 0 1 0, 0 0 1 | −E | 0 −1 0, 0 0 −1 |
| 0 | 0 0 0 | | |

## 3. MHF-PWM Modulation Strategy

There are two low-voltage units with the same DC sides voltage in the asymmetric nine-level inverter, but in the traditional HF-PWM modulation strategy in [18], only one low-voltage unit is PWM-modulated, and the other low-voltage unit is still operated at low frequencies. The modulation strategy also has the problem of unbalanced output power. At low modulation, only some of the units output voltage, the high voltage unit does not participate in the output when the depth of modulation is less than 0.5, and when the modulation depth is lower than 0.25, only the low-voltage unit with PWM modulation is working, thus the output power of each unit is not balance. In the case of high modulation, although each unit has a voltage output, However, the ratio of the amplitude of the fundamental voltage of the three units cannot be maintained at 2:1:1, and there is also a problem that the output power is not balanced. The traditional HF-PWM modulation strategy not only has a large difference in switching times between the two units, but also has an imbalance in output power, and this phenomenon is more serious at low modulation.

If PWM modulation is performed on two low-voltage units at the same time, a carrier phase shift or carrier phase disposition modulation strategy in the cascade modulation strategy can be introduced to change the output characteristics of the two low-voltage units. On this basis, the MHF-PWM modulation strategy is proposed for the traditional HF-PWM modulation low-voltage unit switching loss and output power unbalance. In the MHF-PWM modulation strategy, the high-voltage unit H1 still uses step-wave modulation, which is beneficial to reduce the loss caused by frequent switching of transistors. The low voltage unit adopts the method of PD+PS modulation. PD modulation is used in a single low voltage unit, and PS modulation is used between two low voltage units. The power balance between the two low-voltage units can be automatically realized in the unit output cycle, and the average number of switching times and switching losses of the low-voltage unit can be realized between the two units.

The modulation principle of MHF-PWM modulation strategy is shown in Figure 2. In the figure, from top to bottom are the H1 unit modulation principle, the H1 unit output waveforms $u_{H1}$, H2 and H3 unit modulation principle, the H2 unit output waveform $u_{H2}$, the H3 unit output waveform $u_{H3}$, and the inverter output phase voltage $u_{AN}$ waveform. The modulation wave $v_m$ is a sine wave. When the modulation wave $v_m$ is greater than 2$E$, that is, $v_m$ is greater than $v_{cr1}$, the H1 unit outputs +2$E$; when the modulation wave $v_m$ is less than −2$E$, that is, $v_m$ is less than $v_{cr1−}$, the H1 unit outputs −2$E$, In other cases, the H1 unit does not participate in the output. It can be seen that the output voltage $u_{H1}$ of the H1 unit is a square wave and the unit operates at the fundamental frequency, which can reduce the switching loss of the high voltage unit. H2 and H3 share a modulated wave $v'_m$, which is obtained by subtracting the output voltage $u_{H1}$ of the H1 unit from the modulated wave $v_m$. $v_{cr2}$ and $v_{cr2−}$ are carriers of the H2 unit, and $v_{cr3}$ and $v_{cr3−}$ are carriers of the H3 unit. The amplitude and frequency of the four carriers are the same, and the two complementary carriers in the same unit are arranged in reverse stack. In these four carriers, the phases of $v_{cr2}$ and $v_{cr3}$, $v_{cr2−}$ and $v_{cr3−}$ are 180° different from each other to form a phase shift arrangement. It is possible to make the two units output the fundamental amplitudes the same. According to the PS+PD modulation, the output voltages of the two units are $u_{H2}$ and $u_{H3}$ respectively. It can be seen that the output voltages of the H2 and H3 units are both PWM waves, and the MHF-PWM modulation strategy realizes PWM modulation for the two low-voltage units.

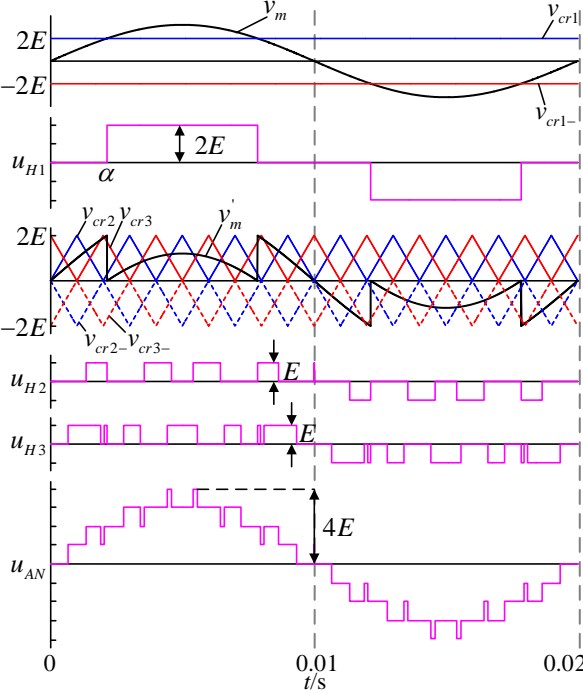

**Figure 2.** Principle of MHF-PWM modulation strategy.

## 4. Analyze of MHF-PWM Modulation Strategy Output Power and Power Balance Method

Let $P_{Hi}$ ($i$ = 1, 2, 3) be the active power output by each of the three units, then the output power of each unit of the asymmetric nine-level inverter in one cycle can be expressed as:

$$P_{Hi} = \frac{1}{T} \int_0^T u_{Hi} i_{Hi} dt \tag{3}$$

where $i_{Hi}$ ($i$ = 1, 2, 3) is the output current of each unit. Since the H-bridge in the asymmetric nine-level inverter topology is cascaded, the output current $i_{Hi}$ of each unit is the same as the inverter output phase current $i_o$. That is, $i_{H1} = i_{H2} = i_{H3} = i_o$, and Equation (3) can be rewritten as:

$$P_{Hi} = \frac{1}{T} \int_0^T u_{Hi} i_o dt \tag{4}$$

The phase current $i_o$ output by the inverter during normal operation is a sine wave, and its output waveform can be expressed as

$$i_o = I_o \sin(\omega t + \varphi) \tag{5}$$

where $i_o$ is the output phase current amplitude, $\omega$ is the output current angular frequency, and $\varphi$ is the phase angle between the output current and the output voltage fundamental wave. According to Equations (4) and (5), $P_{Hi}$ can be expressed as

$$\begin{aligned} P_{Hi} &= \frac{1}{T} \int_0^T u_{Hi} I_o \sin(\omega t + \varphi) dt \\ &= \frac{1}{2} u_{Hi(1)} I_o \cos\varphi \end{aligned} \tag{6}$$

In Equation (6), $u_{Hi(1)}$ ($i$ = 1, 2, 3) is the amplitude of the output voltage fundamental of each unit, and the ratio of the output power of each unit can be expressed as

$$P_{H1} : P_{H2} : P_{H3} = u_{H1(1)} : u_{H2(1)} : u_{H3(1)} \tag{7}$$

According to formula (7), the ratio of the output power of each unit of the asymmetric nine-level inverter is equal to the ratio of the amplitude of the fundamental voltage of each unit, to analyze the relationship between the output power of each unit of the inverter, it is only necessary to analyze the relationship between the amplitudes of the fundamental voltages of each unit.

If the angular frequency of the sinusoidal modulated wave is $\omega$ and the depth of modulation is $m_a$ ($0 \le m_a \le 1$), the sinusoidal modulated wave can be expressed as

$$v_m = 4E m_a \sin(\omega t) \tag{8}$$

Since the high voltage unit H1 adopts step wave modulation, the high voltage unit has an output only when the modulated wave is greater than 2$E$ or less than −2$E$, and the opening angle of the high voltage unit in the positive half cycle is $\alpha$ ($0 \le \alpha \le \pi/2$), The relationship between the opening angle $\alpha$ and the depth of modulation can be obtained by $v_m = 2E$ in Equation (8).

$$\alpha = \arcsin\frac{1}{2m_a}, (0.5 \le m_a \le 1) \tag{9}$$

When $0 \le m_a \le 0.5$, the modulated wave is always between 2$E$ and −2$E$, the high voltage unit does not participate in the voltage output, $\alpha$ is always 90° when 0.5 < ma ≤ 1, as the modulation increases, $\alpha$ from 90° gradually decreases, the high voltage unit outputs the step wave $u_{H1}$, and the general expression of Fourier decomposition is

$$u_{H1} = \frac{a_0}{2} + \sum_{n=1}^{\infty} [a_n \cos(n\omega t) + b_n \sin(n\omega t)] \tag{10}$$

Since $u_{H1}$ is an odd function, $a_0 = 0$, $a_n = 0$, only $b_n$ is not 0, and the expression of $b_n$ is

$$b_n = \frac{4}{T} \int_\alpha^{\pi-\alpha} 2E \sin(n\theta) d\theta, n = 1, 2 \cdots \tag{11}$$

Bringing Equation (11) into Equation (10), and by simplification, one can obtain the Fourier series expression of the output voltage

$$u_{H1} = \frac{8E}{\pi} \sum_{n=1,3,5\cdots}^{\infty} \frac{1}{n} \cos(n\alpha) \sin(n\omega t) \tag{12}$$

Let n = 1 in Equation (12), the fundamental amplitude of the output voltage of the high voltage unit is

$$u_{H1(1)} = \frac{8E}{\pi} \cos \alpha \tag{13}$$

Combining Equations (9) and (13), we can obtain the relationship between $u_{H1(1)}$ and the modulation depth.

$$u_{H1(1)} = \frac{8E}{\pi} \sqrt{1 - \frac{1}{4m_a^2}} \tag{14}$$

Based on Equation (14), at full modulation, $u_{H1}$ is

$$u_{H1(1)} = \begin{cases} 0 & m_a \in [0, 0.5] \\ \frac{8E}{\pi} \sqrt{1 - \frac{1}{4m_a^2}} & m_a \in (0.5, 1] \end{cases} \tag{15}$$

The modulated wave $v'_m$ of the low voltage unit is obtained by subtracting $u_{H1}$ of the H1 unit from $v_m$

$$v'_m = v_m - u_{H1} \tag{16}$$

The two low-voltage units share a modulated wave and are modulated by the PS+PD method. The modulation method can make the output voltages of the two low-voltage units have the same amplitude and each one output half of the amplitude of $v'_m$. According to Equations (15) and (16), at full modulation, $u_{H2(1)}$ and $u_{H3(1)}$ is

$$u_{H2(1)} = u_{H3(1)} = \frac{1}{2}(4Em_a - u_{H1(1)}) = \begin{cases} 2Em_a & m_a \in [0, 0.5] \\ 2Em_a - \frac{4E}{\pi} \sqrt{1 - \frac{1}{4m_a^2}} & m_a \in (0.5, 1] \end{cases} \tag{17}$$

According to Equations (15) and (17), it can be seen that the MHF-PWM modulation strategy can only guarantee the output power balance between H2 and H3, and the output power balance between the three units cannot be achieved.

According to formula (13), $u_{H1(1)}$ is related to the conduction angle $\alpha$, by changing the conduction angle of the high voltage unit, the amplitude of the fundamental voltage of the output voltage of the high voltage unit can be controlled. If the amplitude of the output voltage of the high voltage unit is always 1/2 of the amplitude of the modulation wave, the two low voltage units jointly output the remaining 1/2 fundamental amplitude, under the PS+PD modulation strategy, the output voltages of the two low-voltage units are the same and output 1/4 of the fundamental amplitude. According to this principle, the output power balance control of the three units of the asymmetric nine-level inverter under the MHF-PWM modulation strategy can be realized.

Let $u_{H1(1)}$ always be 1/2 modulated wave amplitude

$$u_{H1(1)} = \frac{8E}{\pi} \cos \alpha = 2Em_a \tag{18}$$

Solving the relationship between the conduction angle $\alpha$ of the high voltage unit and the modulation depth $m_a$ under full modulation

$$\alpha = \arccos(\frac{\pi m_a}{4}) \tag{19}$$

The angle obtained by formula (19) will be used to control the turning on and off of the high voltage unit. The fundamental amplitude of the low-voltage units' output is

$$u_{H2(1)} = u_{H3(1)} = \frac{1}{2}\left(4Em_a - u_{H1(1)}\right)$$
$$= Em_a \tag{20}$$

According to Equations (18) and (20), the power balance MHF-PWM modulation strategy enables high voltage units to participate in voltage output at full modulation, and by controlling the turn-on angle $\alpha$, the output voltage fundamental amplitude of the high voltage unit is always half of the total output fundamental amplitude. The modulation strategy also keeps the amplitudes of the output voltages of the two low-voltage units equal. Finally, the ratio of the output voltage fundamental amplitude of the three units can be controlled to 2:1:1 at full modulation depth to achieve output power balance.

## 5. Simulation

In order to verify the correctness of the proposed power-balanced MHF-PWM modulation strategy, the simulation model of MHF-PWM modulation strategy and its power-balance modulation method is built in MATLAB/simulink respectively. Simulation parameters: Unit DC input voltage $E$ = 50 V, load $R$ = 20 $\Omega$, and $L$ = 4 mH. The frequency of sinusoidal modulated wave is 50Hz, the triangular carrier frequency of MHF-PWM and power-balanced MHF-PWM modulation strategies is 2 kHz, and the traditional HF-PWM is 4 kHz.

Figure 3a illustrates the relationship between the fundamental wave of the output voltage of each unit and the modulation depth obtained by the traditional modulation strategy in [18]. The fundamental amplitude of the three elements cannot be linearly related to the modulation depth. According to the analysis of Equation (7), it can be seen that this modulation strategy cannot achieve power balancing. Figure 3b shows the relationship between the amplitude of the fundamental voltage of the output voltage and the modulation depth of the three units at full modulation. It can be seen that $u_{H1(1)} : u_{H2(1)} : u_{H3(1)}$ = 2:1:1 is only available when the modulation depth is $m_a$ = 0.556, which achieves output power balance between three units, under the other modulation depths, the amplitude of the fundamental voltage of the three units cannot satisfy this relationship.

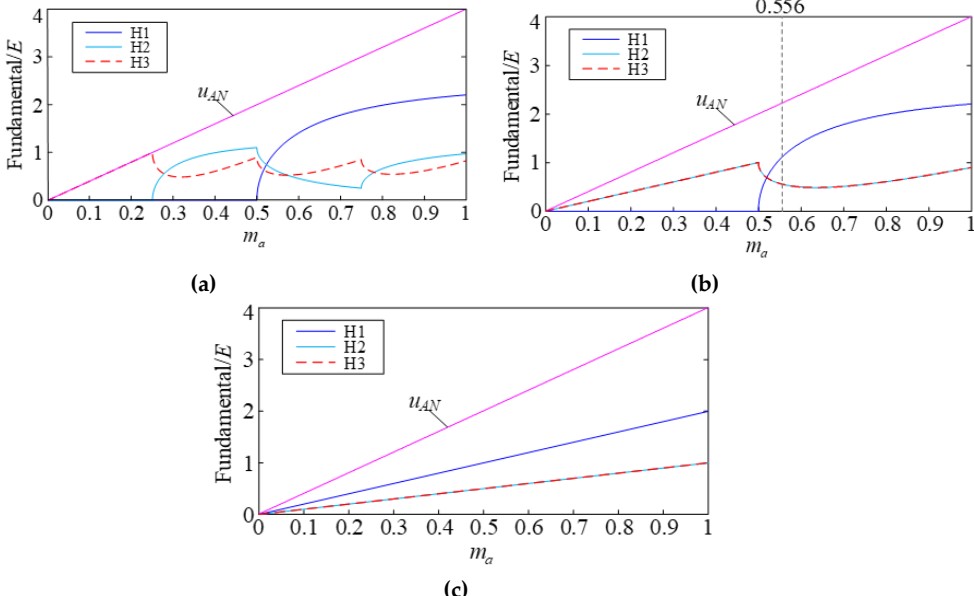

**Figure 3.** Relationship between the amplitude of the fundamental wave and the modulation depth of each unit. (**a**) Traditional HF-PWM. (**b**) MHF-PWM. (**c**) Power balance MHF-PWM.

It can be seen that the MHF-PWM modulation strategy cannot achieve output power balance at all modulation depth. It can be seen from Figure 3c that there is a linear relationship between the amplitude of the three unit output fundamentals and the modulation depth. The power balance MHF-PWM modulation strategy can make the output of three units have $u_{H1(1)}: u_{H2(1)}: u_{H3(1)} = 2:1:1$ at any modulation depth, this shows that this power balance modulation strategy achieves output power balance between the three units.

Figure 4a is the output voltage waveform of each unit under the traditional HF-PWM modulation strategy. It can be seen that units H1 and H2 work at low frequency and unit H3 works at high frequency. The switching frequency between two low-voltage units is extremely different.

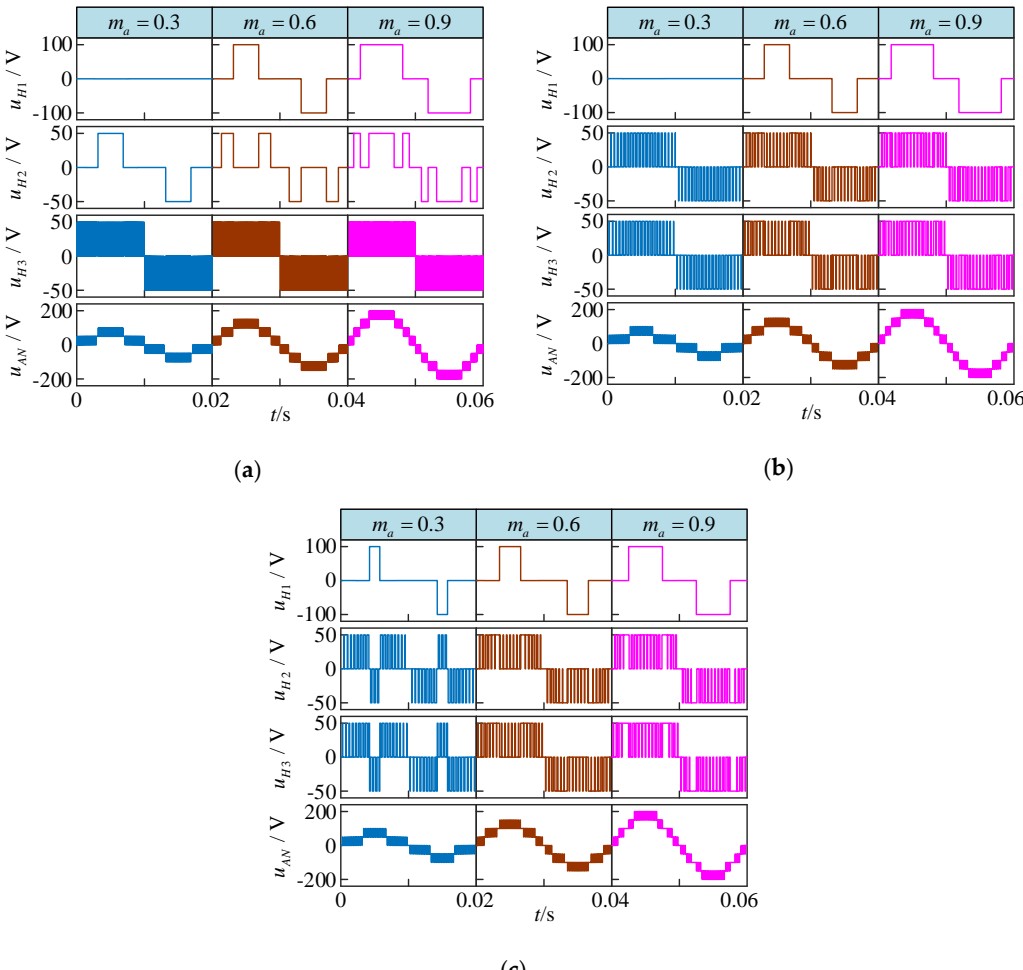

**Figure 4.** Output voltage waveforms of two modulation strategies when the $m_a$ = 0.3, 0.6, 0.9. (**a**) Traditional HF-PWM. (**b**) MHF-PWM. (**c**)Power balance MHF-PWM.

Figure 4b shows the waveforms of the output voltage and output phase voltage of each unit in the MHF-PWM modulation strategy at three modulation depths of $m_a$ = 0.3, 0.6, and 0.9. It can be seen that the high voltage unit does not participate in the voltage output at low modulation, the output voltage waveforms of the two low-voltage units are the same under each modulation depth. Symmetrical output phase voltage can be obtained by superimposing the three cell output voltages.

It can be seen from Figure 4c that the three units have voltage outputs at each modulation depth, and the output voltage waveforms of the two low voltage units are also identical. Comparing Figure 4a–c, although the output voltage waveforms of the corresponding units of the two modulation strategies are different, PWM waves of the same level can be output when the modulation depths are the same.

As shown in Figure 5a, under the three modulation depths of ma = 0.3, 0.6, and 0.9, there is also a gap in the output power between the two low-voltage units, and the high-voltage units do not output power at low modulation depths. This shows that the traditional HF-PWM modulation strategy cannot make the output power of each unit meet the relationship of $P_{H1} : P_{H2} : P_{H3}$ = 2:1:1, and cannot achieve power balance.

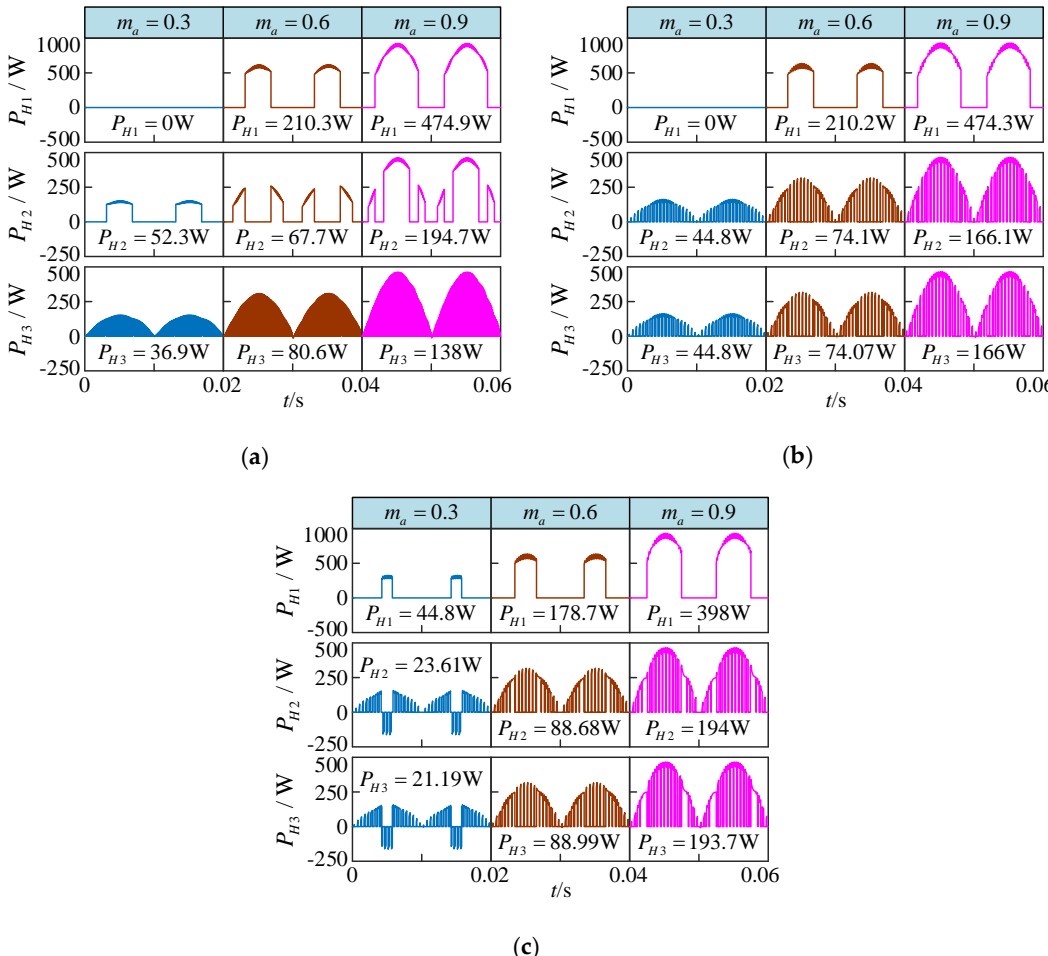

**Figure 5.** Output power waveforms of two modulation strategies when the $m_a$ = 0.3, 0.6, 0.9. (**a**) Traditional HF-PWM. (**b**) MHF-PWM. (**c**)Power balance MHF-PWM.

Figure 5b shows the output power waveform of each output unit for the MHF-PWM modulation strategy at three modulation depths of $m_a$ = 0.3, 0.6, and 0.9. It can be seen from the figure that under the modulation strategy, the low-voltage units H2 and H3 have power output and are basically consistent in the full modulation depth, the high voltage unit only has power output at high modulation levels. Under these three modulation depths, the output power of three cascaded units cannot meet the relationship of $P_{H1} : P_{H2} : P_{H3}$ = 2:1:1. Accordingly, output power balance cannot be achieved between the three units.

Figure 5c shows the output power waveform of each output unit of the power balance MHF-PWM modulation strategy. It can be seen from the figure that H1 always has power output under each modulation depth. The output power of the low-voltage unit H2 and H3 is basically the same, and the output power of the unit H1 is approximately twice the output power of the two low-voltage units. The power balance modulation strategy can make the output power of the three units approximately satisfy the relationship of $P_{H1} : P_{H2} : P_{H3}$ = 2:1:1. The output power balance control between the output units is realized.

Figure 6 represents the spectral distribution diagrams of the output phase voltage $u_{AN}$ for three modulation strategies at three modulation depths of $m_a = 0.3$, 0.6, and 0.9, respectively. The harmonic distribution of traditional HF-PWM and MHF-PWM is different under the same modulation depth, which is mainly caused by the difference between the two modulation strategies, otherwise, the total harmonic distortion (THD) of the two methods are exactly the same. When $m_a = 0.3$ and 0.6, the spectral distribution and THD of the phase voltages of the MHF-PWM and power balance MHF-PWM modulation strategies is basically the same. When $m_a = 0.9$, we can see that the power balance MHF-PWM modulation strategy output phase voltage $u_{AN}$ spectrum diagram is basically the same as the MHF-PWM modulation strategy except for some low-order harmonics. Moreover, the power balanced MHF-PWM modulation strategy output phase voltage THD is slightly reduced compared to the traditional HF-PWM and MHF-PWM.

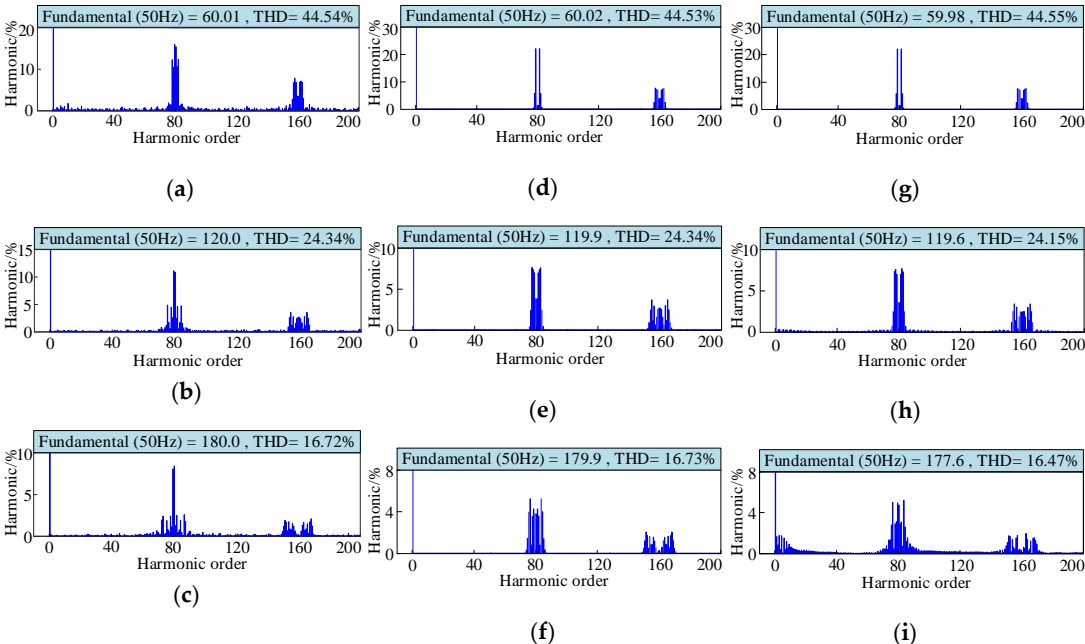

**Figure 6.** Output phase voltage $u_{AN}$ spectrum at different modulation depths. Traditional HF-PWM: (**a**) $m_a = 0.3$; (**b**) $m_a = 0.6$; (**c**) $m_a = 0.9$. MHF-PWM: (**d**) $m_a = 0.3$; (**e**) $m_a = 0.6$; (**f**) $m_a = 0.9$. Power balance MHF-PWM: (**g**) ma = 0.3; (**h**) ma = 0.6; (**i**) ma = 0.9.

Figure 7 is a graph showing the relationship between the output phase voltage THD and the modulation depth $m_a$ of the three modulation strategies in the modulation depth of 0.1–1. It can be seen that the THD of traditional HF-PWM and MHF-PWM is the same in this modulation depth range. The THD of the power balance MHF-PWM modulation strategy is the same as the traditional HF-PWM and MHF-PWM modulation strategy with a modulation depth of 0.1–0.6, the THD of the output phase voltage of the power balance MHF-PWM modulation strategy is slightly better than the other two modulation strategies in the modulation depth of 0.6–1. This shows that the power balance MHF-PWM modulation strategy does not degrade the output power quality.

In summary, the power balance MHF-PWM modulation strategy can achieve output power balance between the three units without affecting the output power quality.

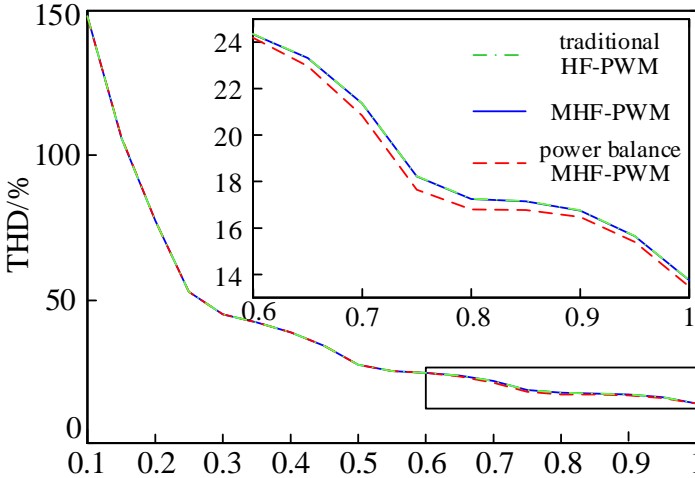

**Figure 7.** The relationship between the output voltage THD and the modulation depth of three modulation strategies.

## 6. Experiment

In order to verify the feasibility of the proposed modulation strategy, a power balanced MHF-PWM modulation strategy experimental platform was built. The experimental platform is controlled by DSP. The DC side voltage of the H-bridge is set to 50 V, 50 V, 100 V, the load resistance is 20 Ω, and the filter inductance is 4 mH. The carrier frequency is 5 kHz and the modulation wave frequency is 50 Hz. The experiments were carried out at a modulation depth of 0.6 and 0.9, experimental prototype is shown in Figure 8 and the experimental results are as follows.

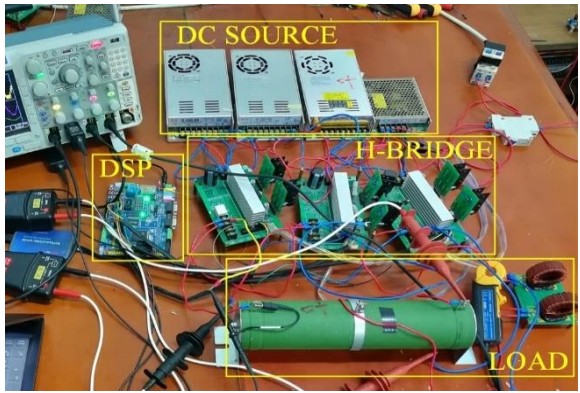

**Figure 8.** Experimental prototype.

It can be seen from Figure 9a that two low-voltage units output PWM waves, and the high-voltage unit outputs stepped waves, The phase voltage waveform obtained by superimposing the output voltages of the three cells is a nine-level PWM wave. Figure 9b shows that the filtered phase voltage waveform is close to the standard sine wave, output phase current is a standard sine wave. This shows that the phase voltage output by the modulation method has better harmonic characteristics. The phase voltage harmonics are mainly distributed at and around 10 kHz, and some low-frequency harmonics are also present at low frequencies in Figure 9c, which is consistent with the previous simulation results.

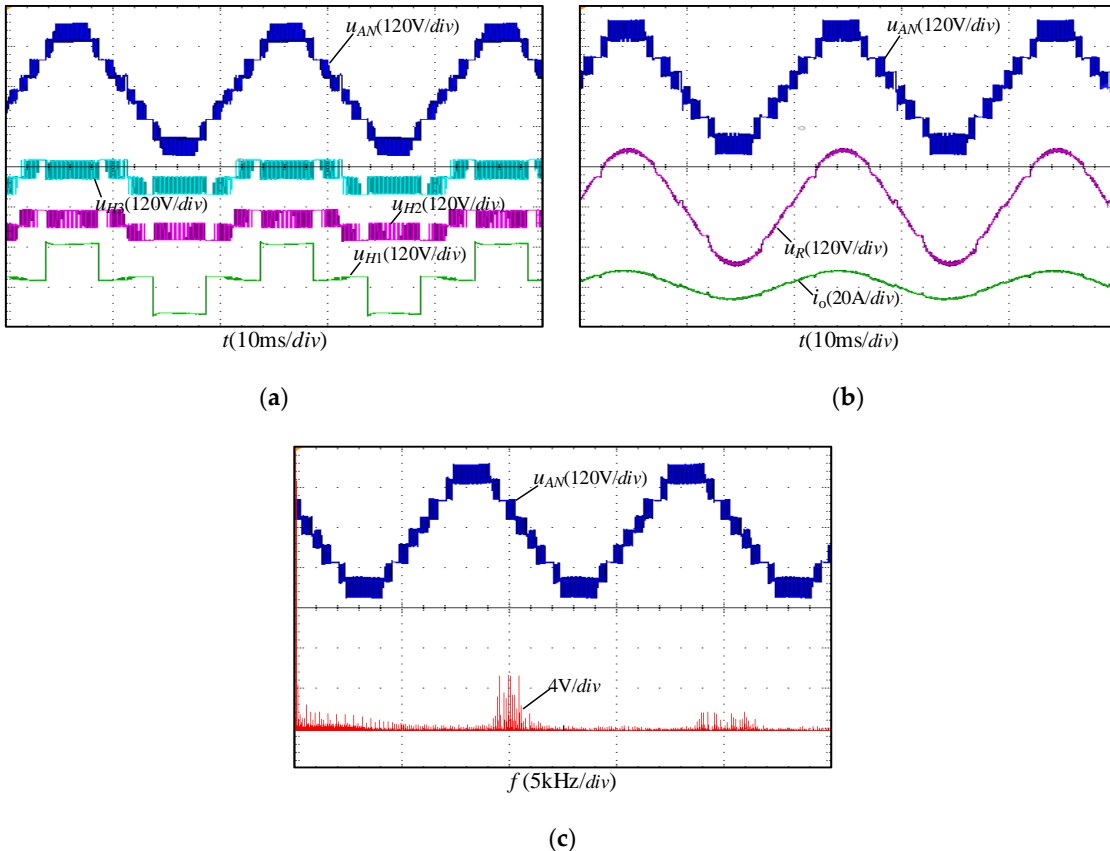

**Figure 9.** Power balanced MHF-PWM strategy output ($m_a$ = 0.9). (**a**) Output voltage and phase voltage waveform of each unit. (**b**) Phase voltage filtered phase voltage and phase current waveform. (**c**) Phase voltage spectrum.

Figure 10 shows the output voltage, phase current and power waveform of each unit when the modulation depth is 0.9. The output power waveform is obtained by multiplying the output voltage of each unit by the phase current. It can be seen that the output waveforms of the two low-voltage units are basically the same, and the output power of the two is also very close. The average power output of the three units is 394.3 W, 192.2 W, and 191.8 W, respectively. Based on the output power of the H3 unit, the ratio of the output power of each unit is 2.05:1:1, which is very close to 2:1:1. This indicates that the power balance strategy proposed in this paper has a good control effect on the output power of each unit.

Figure 11 shows the output voltage current waveform and phase voltage spectrum of the power balance modulation strategy when the modulation depth is 0.6. It can be seen from Figure 11a that the output waveforms of the two low-voltage units are also PWM waves, and the high-voltage unit output step waves, compared with the modulation depth of 0.9, the output step wave width is significantly reduced. The phase voltage waveform obtained by superimposing the output voltage of each unit is a seven-level PWM wave. It can be seen from Figure 11b that the phase voltage after inductive filtering is also very close to a sine wave. As can be seen from the output phase voltage spectrum diagram in Figure 11c, its harmonics are also distributed at 10 kHz and its vicinity, and there are also a few harmonics at low frequencies, which is consistent with the simulation results.

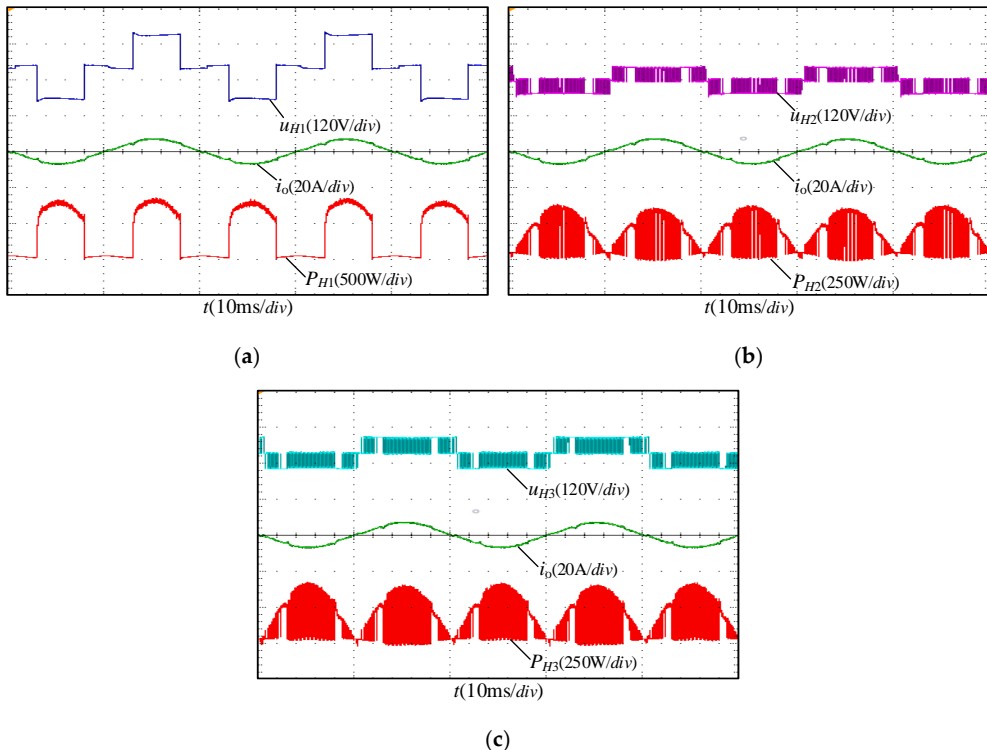

**Figure 10.** Output power of each unit under power balance MHF-PWM modulation strategy ($m_a$ = 0.9). (**a**) H1 unit. (**b**) H2 unit. (**c**) H3 unit.

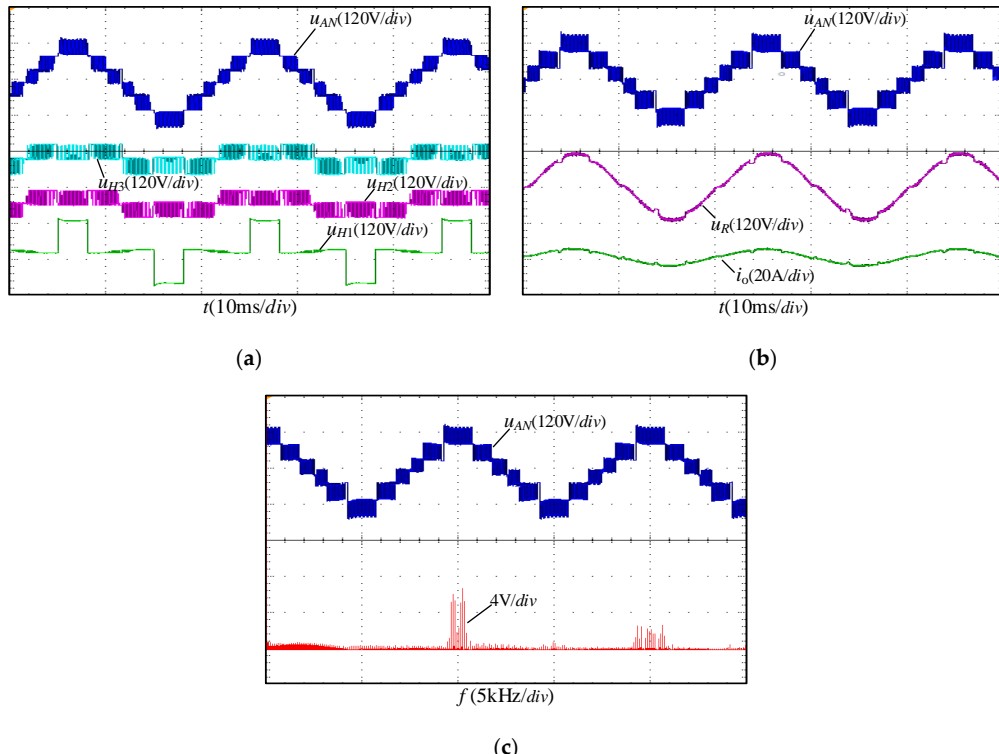

**Figure 11.** Power balanced MHF-PWM strategy output (ma = 0.6). (**a**) Output voltage and phase voltage waveform of each unit. (**b**) Phase voltage filtered phase voltage and phase current waveform. (**c**) Phase voltage spectrum.

Figure 12 shows the output voltage, phase current and power waveform of each unit when the modulation depth is 0.6, it can be seen that the output power of the low voltage unit is still basically the same. It can be seen from the figure that the average output power of each unit is 177.2 W, 87.7 W, 88.2 W, and the ratio of the output power of the three units is 2.01: 1: 1, which means that power balance control is also achieved under this modulation depth.

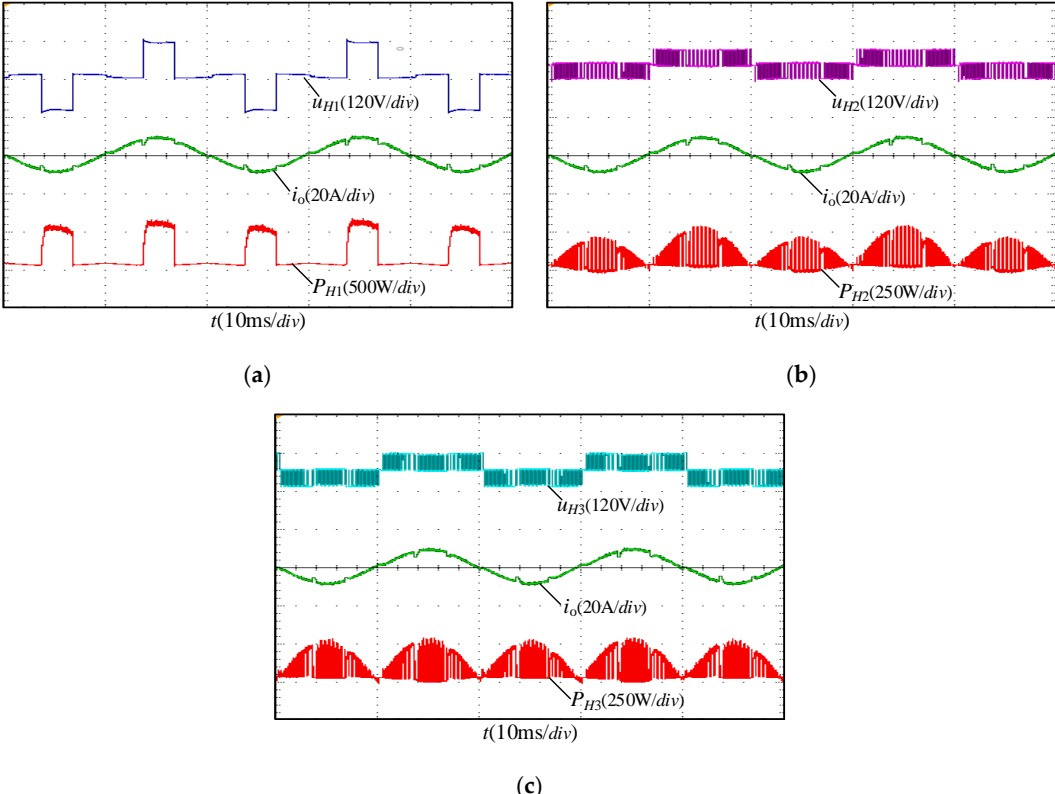

(**a**)    (**b**)

(**c**)

**Figure 12.** Output power of each unit under power balance MHF-PWM modulation strategy ($m_a = 0.6$).
(**a**) H1 unit. (**b**) H2 unit. (**c**) H3 unit.

The above experimental results show that the power balanced MHF-PWM modulation strategy can achieve an equalized distribution of output power between three cascaded units within the full modulation depth.

## 7. Conclusions

Compared with 1:1:1 topology, 1:1:2 topology can output more voltage levels without increasing the number of switches, at the same time, the volume can be kept constant or only slightly increased. Although the traditional HF-PWM modulation strategy has better output harmonic characteristics, it has the problem of unbalanced output power of each unit. The power balanced MHF-PWM modulation strategy proposed in this paper can achieve balanced distribution of inverter output power in all modulation depths. Final theoretical analysis, simulation results, and experimental results are as follows.

(1) The MHF-PWM modulation strategy uses PWM modulation for both low-voltage units at the same time, which can achieve the output power balance and the same switching loss of the two low-voltage units while maintaining the equivalent switching frequency and the output power quality. This is beneficial to solve the problem of uneven temperature rise of the two low-voltage units. Therefore, two low-voltage units can use the same cooling module, reducing procurement and maintenance costs.

(2) Power balance MHF-PWM modulation strategy keeps the ratio of the output power of the three units always at 2:1:1 by limiting the amplitude of the output voltage fundamental of the high voltage unit. This achieves an equalized distribution of the inverter output power between the three cascaded units at all modulation depths. Moreover, the phase voltage harmonic content of output at a higher modulation depth is slightly reduced relative to the MHF-PWM modulation strategy, thereby improving the output power quality.

**Author Contributions:** Project administration, M.Y.; conceptualization, Q.W.; methodology, Q.W., M.Y.; formal analysis, G.S.; software, W.R.; validation, Q.W., W.R. and G.S.; writing—original draft preparation, Q.W.; writing—review and editing, M.Y. All authors have read and agreed to the published version of the manuscript.

**Funding:** This research was funded by National Natural Science Foundation of China, grant number 51767007; Jiangxi Provincial Industrial Science and Technology Support Project, grant number 20192BBEL50011; Jiangxi Natural Science Foundation Project, grant number 20192BAB206036; Jiangxi Provincial Youth Science Foundation of China, grant number GJJ180306.

**Conflicts of Interest:** There are no conflict of interest regarding the publication of this paper.

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
