# Peer review of "Modified Modulation Strategy of 1: 1: 2 Asymmetric Nine-Level Inverter and Its Power Balance Method"

_electronics, doi:10.3390/electronics9010075_

Round 1
Reviewer 1 Report
This paper proposes the Modified Modulation Strategy and Power Balance Control Method for Asymmetric Cascaded Nine-Level Inverters. By using the power balance MHF-PWM modulation strategy in the asymmetric cascaded nine-level inverters, the problem of switching loss and the unbalanced output power is improved for the changes in stage which is compared with the conventional HF-PWM modulation strategy. However, the content of the manuscript still needs to be further clarified. Please address the following points in a revised submission.
(1). Page 1, Line 23-23, 291-292
In general, the cascaded multi-level inverter is widely used in the medium voltage high power transmission system. However, the output power of the asymmetric cascaded nine-level inverters doesn’t seem to have had high power level.
(2). Page 2. Line 43-45
The three-unit cascaded seven-level inverter with the DC voltage ratio 1:1:1 can be implemented in the literature 16. Does the power balance MHF-PWM modulation strategy also can apply in the three-unit cascade seven-level inverter with the same dc voltage ratio 1:1:1?
(3). Page 5, Line 161-162; Page 6, Line 183
The authors need to check the English grammar of manuscript.
Reviewer 2 Report
In this paper, the authors propose a modulation strategy for asymmetric cascade nine-level inverter to realize balanced power dissipations on the two low-voltage H-bridge units. A solid background is provided, and the results are supported by a comprehensive study, including theoretical study, simulation and a hardware experiment. The presentation is well done. Generally, I think it is a good paper with only a few issues needed be addressed:
I would suggest the authors to use a different title, which is currently “Research on …” and is in generalities. In my opinion, the title should have more emphasis on the novelty of the work.
Section 5 demonstrates the simulation results, including the output voltage in Fig. 4, power in Fig.5 and spectral distributions in Fig. 6. However, these results only consider the proposed MHF-PWM and the power balanced strategies. The comparison with state-of-the-art designs is missing. As several previous works ([16-18]) have been described in Section 1, I would like to see a comparison between the proposed design and the previous ones, in terms of the switching loss, percentage THD, range of the modulation depths, etc.
Reviewer 3 Report
Dear Authors, please find my questions and suggestions below.
In lines 83, 84 you wrote: "It can be seen that the inverter has redundant switching states at the outputs ±E, ±2E, ±3E." Why is this clarification here? Does this attribute of function bring any benefits or problems? Please explain it.
You show experimental results at a modulation depth of 0.9. But you don't report about experiments at 0.3 and 0.6 depth modulation. If you have performed experiments for these cases, please add a short description of the results. It may be interesting to readers.
In the conclusion section please discuss more benefits of using your
strategy and method for creating cascaded multi-level inverters. How will the cooling system for such inverters change? Will your approaches reduce the size of the device whit inverters?
There are no references to equations (2), (14), (16), (18), (19), (20) in text. Please fix it.
There is a typo in formulas (15), (17): ma∈(0.5, 1]. Please fix it.
Please check spelling, punctuation, and grammar of the manuscript. I noticed:
line 2
There is space in the "Modulation" word.
line 14
... the same time, output power of conventional modulation strategy is unbalanced.
line 25
...higher number of levels without increasing the withstand voltage of the device. [1-2]. Cascaded
line 30
...Compared with the type I topology, the type II and type III can ...
line 58
...two units, so that the low-voltage unit ...
line 154
... α is always 90°, when 0.5 < ma ≤ 1, as the modulation...
line 201
According to the analysis of equation (7), it can be seen that...
line 217
A Symmetrical output phase voltage can be obtained by superimposing...
line 284
some low-frequency harmonics are also present ...
